# The limits of squared Euclidean distance regularization[*]

**Michał Dereziński**
Computer Science Department
University of California, Santa Cruz
CA 95064, U.S.A.
mderezin@soe.ucsc.edu

**Manfred K. Warmuth**
Computer Science Department
University of California, Santa Cruz
CA 95064, U.S.A.
manfred@cse.ucsc.edu

## Abstract

Some of the simplest loss functions considered in Machine Learning are the square loss, the logistic loss and the hinge loss. The most common family of algorithms, including Gradient Descent (GD) with and without Weight Decay, always predict with a linear combination of the past instances. We give a random construction for sets of examples where the target linear weight vector is trivial to learn but any algorithm from the above family is drastically sub-optimal. Our lower bound on the latter algorithms holds even if the algorithms are enhanced with an arbitrary kernel function.

This type of result was known for the square loss. However, we develop new techniques that let us prove such hardness results for any loss function satisfying some minimal requirements on the loss function (including the three listed above). We also show that algorithms that regularize with the squared Euclidean distance are easily confused by random features. Finally, we conclude by discussing related open problems regarding feed forward neural networks. We conjecture that our hardness results hold for any training algorithm that is based on the squared Euclidean distance regularization (i.e. Back-propagation with the Weight Decay heuristic).

## 1 Introduction

We define a set of simple linear learning problems described by an $n$ dimensional square matrix $\mathbf{M}$ with $\pm 1$ entries. The rows $\mathbf{x}_i$ of $\mathbf{M}$ are $n$ instances, the columns correspond to the $n$ possible targets, and $M_{ij}$ is the label given by target $j$ to the instance $\mathbf{x}_i$ (See Figure 1). Note, that $M_{ij} = \mathbf{x}_i \cdot \mathbf{e}_j$, where $\mathbf{e}_j$ is the $j$-th unit vector. That is, the $j$-th target is a linear function that picks the $j$-th column out of $\mathbf{M}$. It is important to understand that the matrix $\mathbf{M}$, which we call the *problem matrix*, specifies $n$ learning problems: In the $j$th problem each of the $n$ instances (rows) are labeled by the $j$th target (column). The rationale for defining a set of problems instead of a single problem follows from the fact that learning a single problem is easy and we need to average the prediction loss over the $n$ problems to obtain a hardness result.

$$
\begin{array}{cccccc}
 & \rightarrow & -1 & +1 & -1 & +1 \\
instances & \rightarrow & -1 & +1 & +1 & -1 \\
 & \rightarrow & +1 & -1 & -1 & +1 \\
 & \rightarrow & +1 & +1 & -1 & +1 \\
 & & \uparrow & \uparrow & \uparrow & \uparrow \\
 & & & targets & &
\end{array}
$$

**Figure 1:** A random $\pm 1$ matrix $\mathbf{M}$: the instances are the rows and the targets the columns of the matrix. When the $j$-th column is the target, then we have a linear learning problem where the $j$-th unit vector is the target weight vector.

---
[*]This research was supported by the NSF grant IIS-1118028.

The protocol of learning is simple: The algorithm is given $k$ training instances labeled by one of the targets. It then produces a linear weight vector $\mathbf{w}$ that aims to incur small average loss on all $n$ instances labeled by the same target.[1] Any loss function satisfying some minimal assumptions can be used, including the square, the logistic and the hinge loss. We will show that when $\mathbf{M}$ is random, then this type of problems are *hard to learn* by any algorithm from a *certain class of algorithms*.[2]

By hard to learn we mean that the loss is high when we average over instances and targets. The class of algorithms for which we prove our hardness results is any algorithm whose prediction on a new instance vector $\mathbf{x}$ is a function of $\mathbf{w} \cdot \mathbf{x}$ where the weight vector $\mathbf{w}$ is a linear combination of training examples. This includes any algorithm motivated by regularizing with $|| \mathbf{w} ||_2^2$ (i.e. algorithms motivated by the Representer Theorem [KW71, SHS01]) or alternatively any algorithm that exhibits certain rotation invariance properties [WV05, Ng04, WKZ14]. Note that any version of Gradient Descent or Weight Decay on the three loss functions listed above belongs to this class of algorithms, i.e. it predicts with a linear combination of the instances seen so far.

This class of simple algorithms has many advantages (such as the fact that it can be kernelized). However, we show that this class is very slow at learning the simple learning problems described above. More precisely, our lower bounds for a randomly chosen $\mathbf{M}$ have the following form: For some constants $A \in (0, 1]$ and $B \geq 1$ that depend on the loss function, any algorithm that predicts with linear combinations of $k$ instances has average loss at least $A - B\frac{k}{n}$ with high probability, where the average is over instances and targets. This means that after seeing a fraction of $\frac{A}{2B}$ of all $n$ instances, the average loss is still at least the constant $\frac{A}{2}$ (see the red solid curve in Figure 2 for a typical plot of the average loss of GD).

Note, that there are trivial algorithms that learn our learning problem much faster. These algorithms clearly do not predict with a linear combination of the given instances. For example, one simple algorithm keeps track of the set of targets that are consistent with the $k$ examples seen so far (the version space) and chooses one target in the version space at random. This algorithm has the following properties: After seeing $k$ instances, the expected size of the version space is $\min(n/2^k, 1)$, so after $O(\log_2 n)$ examples, with high probability there is only one unit vector $\mathbf{e}_j$ left in the version space that labels all the examples correctly.

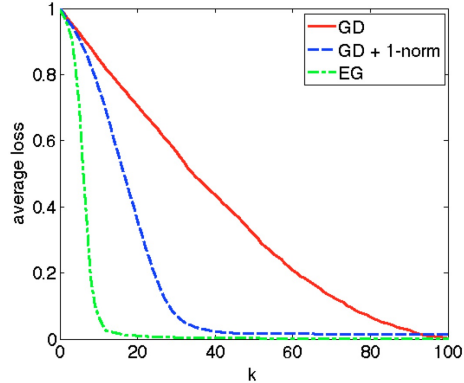

**Figure 2:** The average logistic loss of the Gradient Descent (with and without 1-norm regularization) and the Exponentiated Gradient algorithms for the problem of learning the first column of a 100 dimensional square $\pm 1$ matrix. The $x$-axis is the number of examples $k$ in the training set. Note that the average logistic loss for Gradient Descent decreases roughly linearly.

One way to closely approximate the above version space algorithm is to run the Exponentiated Gradient (EG) algorithm [KW97b] with a large learning rate. The EG algorithm maintains a weight vector which is a probability vector. It updates the weights by multiplying them by non-negative factors and then re-normalizes them to a probability vector. The factors are the exponentiated negative scaled derivatives of the loss. See dot-dashed green curve of Figure 2 for a typical plot of the average loss of EG. It converges "exponentially faster" than GD for the problem given in Figure 1. General regret bounds for the EG algorithm are known (see e.g. [KW97b, HKW99]) that grow logarithmically with the dimension $n$ of the problem. Curiously enough, for the EG family of algorithms, the componentwise logarithm of the weight vector is a linear combination of the instances.[3] If we add a 1-norm regularization to the loss, then GD behaves more like the EG algorithm (see dashed blue curve of Figure 2). In Figure 3 we plot the weights of the EG and GD algorithms (with optimized learning rates) when the target is the first column of a 100 dimensional random matrix.

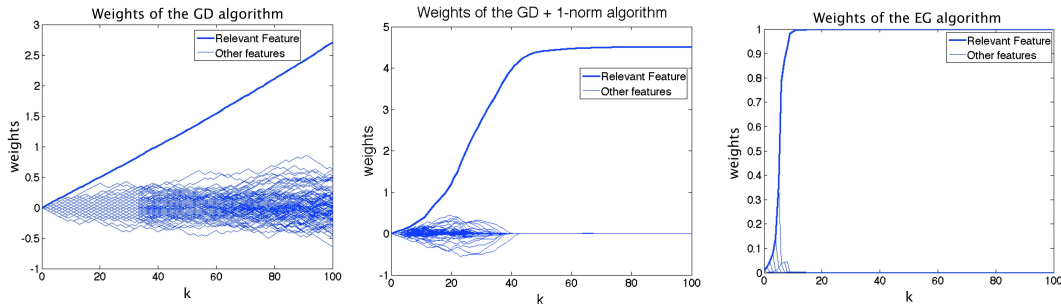

Figure 3: In the learning problem the rows of a 100-dimensional random $\pm 1$ matrix are labeled by the first column. The $x$-axis is the number of instances $k \in 1..100$ seen by the algorithm. We plot all 100 weights of the GD algorithm (left), GD with 1-norm regularization (center) and the EG algorithm (right) as a function of $k$. The GD algorithms keeps lots of small weights around and the first weight grows only linearly. The EG algorithm wipes out the irrelevant weights much faster and brings up the good weight exponentially fast. GD with 1-norm regularization behaves like GD for small $k$ and like EG for large $k$.

The GD algorithm keeps all the small weight around and the weight of the first component only grows linearly. In contrast, the EG algorithm grows the target weight much faster. This is because in a GD algorithm the squared 2-norm regularization does not punish small weight enough (because $w_i^2 \approx 0$ when $w_i$ is small). If we add a 1-norm regularization to the loss then the irrelevant weights of GD disappear more quickly and the algorithm behaves more like EG.

**Kernelization**

We clearly have a simple linear learning problem in Figure 1. So, can we help the class of algorithms that predicts with linear combinations of the instances by "expanding" the instances with a feature map? In other words, we could replace the instance $\mathbf{x}$ by $\phi(\mathbf{x})$, where $\phi$ is any mapping from $\mathbb{R}^n$ to $\mathbb{R}^m$, and $m$ might be much larger than $n$ (and can even be infinite dimensional). The weight vector is now a linear combination of the expanded instances and computing the dot product of this weight vector with a new expanded instance requires the computation of dot products between expanded instances.[4]

Even though the class of algorithms that predicts with a linear combination of instances is good at incorporating such an expansion (also referred to as an embedding into a feature space), we can show that our hardness results still hold even if any such expansion is used. In other words it does not help if the instances (rows) are represented by any other set of vectors in $\mathbb{R}^m$. Note that the learner knows that it will receive examples from one of the $n$ problems specified by the problem matrix $\mathbf{M}$. The expansion is allowed to depend on $\mathbf{M}$, but it has to be chosen before any examples are seen by the learner.

**Related work**

There is a long history for proving hardness results for the class of algorithms that predict with linear combinations of instances [KW97a, KWA97]. In particular, in [WV05] it was shown for the Hadamard matrix and the square loss, that the average loss is at least $1 - \frac{k}{n}$ even if an arbitrary expansion is used. This means, that if the algorithm is given half of all $n$ instances, its average square loss is still half. The underlying model is a simple linear neuron. It was left as an open problem what happens for example for a sigmoided linear neuron and the logistic loss. Can the hardness result be circumvented by choosing different neuron and loss function? In this paper, we are able to show that this type of hardness results for algorithms that predict with a linear combination of the instances are robust to learning with a rather general class of linear neurons and more general loss functions. The hardness result of [WV05] for the square loss followed from a basic property of the Singular Value Decomposition. However, our hardness results require more complicated counting

techniques. For the more general class of loss functions we consider, the Hadamard matrix actually leads to a weaker bound and we had to use random matrices instead.

Moreover, it was shown experimentally in [WV05] (and to some extent theoretically in [Ng04]) that the generalization bounds of 1-norm regularized linear regression grows logarithmically with the dimension $n$ of the problem. Also, a linear lower bound for any algorithm that predicts with linear combinations of instances was given in Theorem 4.3 of [Ng04]. However, the given lower bound is based on the fact that the Vapnik Chervonienkis (VC) dimension of $n$-dimensional halfspaces is $n + 1$ and the resulting linear lower bound holds for any algorithm. No particular problem is given that is easy to learn by say multiplicative updates and hard to learn by GD. In contrast, we give a random problem in Figure 1 that is trivial to learn by some algorithms, but hard to learn by the natural and most commonly used class of algorithms which predicts with linear combinations of instances. Note, that the number of target concepts we are trying to learn is $n$, and therefore the VC dimension of our problem is at most $\log_2 n$.

There is also a large body of work that shows that certain problems cannot be embedded with a large 2-norm margin (see [FS02, BDES02] and the more recent work on similarity functions [BBS08]). An embedding with large margins allows for good generalization bounds. This means that if a problem cannot be embedded with a large margin, then the generalization bounds based on the margin argument are weak. However we don't know of any hardness results for the family of algorithms that predict with linear combinations in terms of a margin argument, i.e. lower bounds of generalization for this class of algorithms that is based on non-embeddability with large 2-norm margins.

**Random features**

The purpose of this type of research is to delineate which types of problems can or cannot be efficiently learned by certain classes of algorithms. We give a problem for which the sample complexity of the trivial algorithm is logarithmic in $n$, whereas it is linear in $n$ for the natural class of algorithms that predicts with the linear combination of instances. However, why should we consider learning problems that pick columns out of a random matrix? Natural data is never random. However, the problem with this class of algorithms is much more fundamental. We will argue in Section 4 that those algorithms get confused by random irrelevant features. This is a problem if datasets are based on some physical phenomena and that contain at least some random or noisy features. It seems that because of the weak regularization of small weights (i.e. $w_i^2 \approx 0$ when $w_i$ is small), the algorithms are given the freedom to fit noisy features.

**Outline**

After giving some notation in the next section and defining the class of loss functions we consider, we prove our main hardness result in Section 3. We then argue that the family of algorithms that predicts with linear combination of instances gets confused by random features (Section 4). Finally, we conclude by discussing related open problems regarding feed forward neural nets in Section 5: We conjecture that going from single neurons to neural nets does not help as long as the training algorithm is Gradient Descent with a squared Euclidean distance regularization.

## 2  Notations

We will now describe our learning problem and some notations for representing algorithms that predict with a linear combination of instances. Let $\mathbf{M}$ be a $\pm 1$ valued *problem matrix*. For the sake of simplicity we assume $\mathbf{M}$ is square ($n \times n$). The $i$-th row of $\mathbf{M}$ (denoted as $\mathbf{x}_i$) is the $i$-th instance vector, while the $j$-th column of $\mathbf{M}$ is the labeling of the instances by the $j$-th target. We allow the learner to map the instances to an $m$-dimensional feature space, that is, $\mathbf{x}_i$ is replaced by $\phi(\mathbf{x}_i)$, where $\phi : \mathbb{R}^n \to \mathbb{R}^m$ is an arbitrary mapping. We let $\mathbf{Z} \in \mathbb{R}^{n \times m}$ denote the new instance matrix with its $i$-th row being $\phi(\mathbf{x}_i)$.[5]

The algorithm is given the first $k$ rows of $\mathbf{Z}$ labeled by one of the $n$ targets. We use $\widehat{\mathbf{Z}}$ to denote the first $k$ rows of $\mathbf{Z}$. After seeing the rows of $\widehat{\mathbf{Z}}$ labeled by target $i$, the algorithm produces a linear combination $\mathbf{w}_i$ of the $k$ rows. Thus the weight vector $\mathbf{w}_i$ takes the form $\mathbf{w}_i = \widehat{\mathbf{Z}}^\top \mathbf{a}_i$, where $\mathbf{a}_i$ is the vector of the $k$ linear coefficients. We aggregate the $n$ weight vectors and coefficients into the $m \times n$ and $k \times n$ matrices, respectively: $\mathbf{W} := [\mathbf{w}_1, \ldots, \mathbf{w}_n]$ and $\mathbf{A} = [\mathbf{a}_1, \ldots, \mathbf{a}_n]$. Clearly, $\mathbf{W} = \widehat{\mathbf{Z}}^\top \mathbf{A}$. By applying the weight matrix to the instance matrix $\mathbf{Z}$ we can obtain the $n \times n$ prediction matrix of the algorithm: $\mathbf{P} = \mathbf{Z}\,\mathbf{W} = \mathbf{Z}\,\widehat{\mathbf{Z}}^\top \mathbf{A}$. Note that $P_{ij} = \phi(\mathbf{x}_i) \cdot \mathbf{w}_j$ is the linear activation of the algorithm produced for the $i$-th instance after receiving the first $k$ rows of $\mathbf{Z}$ labeled with the $j$-th target.

We are now interested to compare the prediction matrix with the problem matrix using a non-negative loss function $L : \mathbb{R} \times \{-1, 1\} \to \mathbb{R}_{\geq 0}$. We define the average loss of the algorithm as

$$\frac{1}{n^2} \sum_{i,j} L(P_{i,j}, M_{i,j}).$$

Note that the loss is between linear activations and binary labels and we average it over instances and targets.

**Definition 1** *We will call a loss function $L : \mathbb{R} \times \{-1, 1\} \to \mathbb{R}_{\geq 0}$ to be $C$-regular where $C > 0$, if $L(a, y) \geq C$ whenever $a \cdot y \leq 0$, i.e. $a$ and $y$ have different signs.*

The loss function guarantees that if the algorithm produces a linear activation of a different sign, then a loss of at least $C$ is incurred. Three commonly used 1-regular losses are the:

- Square Loss, $L(a, y) = (a - y)^2$, used in Linear Regression.
- Logistic Loss, $L(a, y) = -\frac{y+1}{2} \log_2(\sigma(a)) - \frac{y-1}{2} \log_2(1 - \sigma(a))$, used in Logistic Regression. Here $\sigma(a)$ denotes the sigmoid function $\frac{1}{1 + \exp(-a)}$.
- Hinge Loss, $L(a, y) = \max(0, 1 - ay)$, used in Support Vector Machines.

[WV05] obtained a linear lower bound for the square:

**Theorem 2** *If the problem matrix $\mathbf{M}$ is the $n$ dimensional Hadamard matrix, then for any algorithm that predicts with linear combinations of expanded training instances, the average square loss after observing $k$ instances is at least $1 - \frac{k}{n}$.*

The key observation used in the proof of this theorem is that the prediction matrix $\mathbf{P} = \mathbf{Z}\,\widehat{\mathbf{Z}}^\top \mathbf{A}$ has rank at most $k$, because $\widehat{\mathbf{Z}}$ has only $k$ rows. Using an elementary property of the singular value decomposition, the total squared loss $\|\mathbf{P} - \mathbf{M}\|_2^2$ can be bounded by the sum of the squares of the last $n - k$ singular values of the problem matrix $\mathbf{M}$. The bound now follows from the fact that Hadamard matrices have a flat spectrum. Random matrices have a "flat enough" spectrum and the same technique gives an expected linear lower bound for random problem matrices. Unfortunately the singular value argument only applies to the square loss. For example, for the logistic loss the problem is much different. In that case it would be natural to define the $n \times n$ prediction matrix as $\sigma(\mathbf{Z}\,\mathbf{W}) = \sigma(\mathbf{Z}\,\widehat{\mathbf{Z}}^\top \mathbf{A})$. However the rank of $\sigma(\mathbf{Z}\,\mathbf{W})$ jumps to $n$ even for small values of $k$. Instead we keep the prediction matrix $\mathbf{P}$ as the $n^2$ linear activations $\mathbf{Z}\,\widehat{\mathbf{Z}}^\top \mathbf{A}$ produced by the algorithm, and define the loss between linear activations and labels. This matrix still has rank at most $k$. In the next section, we will use this fact in a counting argument involving the possible sign patterns produced by low rank matrices.

If the algorithms are allowed to start with a non-zero initial weight vector, then the hardness results essentially hold for the class of algorithms that predict with linear combinations of this weight vector and the $k$ expanded training instances. The only difference is that the rank of the prediction matrix is now at most $k+1$ instead of $k$ and therefore the lower bound of the above theorem becomes $1 - \frac{k+1}{n}$ instead of $1 - \frac{k}{n}$. Our main result also relies on the rank of the prediction matrix and therefore it allows for a similar adjustment of the bound when an initial weight vector is used.

## 3 Main Result

In this section we present a new technique for proving lower bounds on the average loss for the sparse learning problem discussed in this paper. The lower bound applies to any regular loss and is based on counting the number of sign-patterns that can be generated by a low-rank matrix. Bounds on the number of such sign patterns were first introduced in [AFR85]. As a corollary of our method, we also obtain a lower bound for the "rigidity" of random matrices.

**Theorem 3** *Let $L$ be a $C$-regular loss function. A random $n \times n$ problem matrix $\mathbf{M}$ almost certainly has the property that for any algorithm that predicts with linear combinations of expanded training instances, the average square loss $L$ after observing $k$ instances is at least $4C\left(\frac{1}{20} - \frac{k}{n}\right)$.*

**Proof** $C$-regular losses are at least $C$ if the sign of the linear activation for an example does not match the label. So, we can focus on counting the number of linear activations that have wrong signs. Let $\mathbf{P}$ be the $n \times n$ prediction matrix after receiving $k$ instances. Furthermore let $\text{sign}(\mathbf{P}) \in \{-1,1\}^{n \times n}$ denote the sign-pattern of $\mathbf{P}$. For the sake of simplicity, we define $\text{sign}(0)$ as 1. This simplification underestimates the number of disagreements. However we still have the property that for any $C$-regular loss: $L(a,y) \geq C|\text{sign}(a) - y|/2$.

We now count the number of entries on which $\text{sign}(\mathbf{P})$ disagrees with $\mathbf{M}$. We use the fact that $\mathbf{P}$ has rank at most $k$. The number of sign patterns of $n \times m$ rank $\leq k$ matrices is bounded as follows (This was essentially shown[6] in [AFR85], the exact bound we use below is a refinement given in [Sre04]):

$$f(n,m,k) \leq \left(\frac{8e \cdot 2 \cdot nm}{k(n+m)}\right)^{k(n+m)}.$$

Setting $n = m = a \cdot k$, we get

$$f(n,n,n/a) \leq 2^{(6+2\log_2(e\cdot a))\cdot n^2/a}.$$

Now, suppose that we allow additional up to $r = \alpha n^2$ signs of $\text{sign}(\mathbf{P})$ to be flipped. In other words, we consider the set $S_n^k(r)$ of sign-patterns having Hamming distance at most $r$ from any sign-pattern produced from a matrix of rank at most $k$. For a fixed sign-pattern, the number $g(n,\alpha)$ of matrices obtained by flipping at most $r$ entries is the number of subsets of size $r$ or less that can be flipped:

$$g(n,\alpha) = \sum_{i=0}^{\alpha n^2} \binom{n^2}{i} \leq 2^{H(\alpha)n^2}.$$

Here, $H$ denotes the binary entropy. The above bound holds for any $\alpha \leq \frac{1}{2}$. Combining the two bounds described above, we can finally estimate the size of $S_n^k(r)$:

$$|S_n^k(r)| \leq f(n,n,n/a) \cdot g(n,\alpha) \leq 2^{(6+2\log_2(e\cdot a))\cdot n^2/a} \cdot 2^{H(\alpha)n^2} = 2^{\left(\frac{6+2\log_2(e\cdot a)}{a} + H(\alpha)\right)n^2}.$$

Notice, that if the problem matrix $\mathbf{M}$ does not belong to $S_n^k(r)$, then our prediction matrix $\mathbf{P}$ will make more than $r$ sign errors. We assumed that $\mathbf{M}$ is selected randomly from the set $\{-1,1\}^{n \times n}$ which contains $2^{n^2}$ elements. From simple asymptotic analysis, we can conclude that for large enough $n$, the set $S_n^k(r)$ will be much smaller than $\{-1,1\}^{n \times n}$, if the following condition holds:

$$\frac{6 + 2\log_2(e \cdot a)}{a} + H(\alpha) \leq 1 - \delta < 1. \tag{1}$$

In that case, the probability of a random problem matrix belonging to $S_n^k(r)$ is at most

$$\frac{2^{(1-\delta)n^2}}{2^{n^2}} = 2^{-\delta n^2} \longrightarrow 0.$$

We can numerically solve Inequality (1) for $\alpha$ by comparing the left-hand side expression to 1. Figure 4 shows the plot of $\alpha$ against the value of $\frac{k}{n} = a^{-1}$. From this, we can obtain the simple

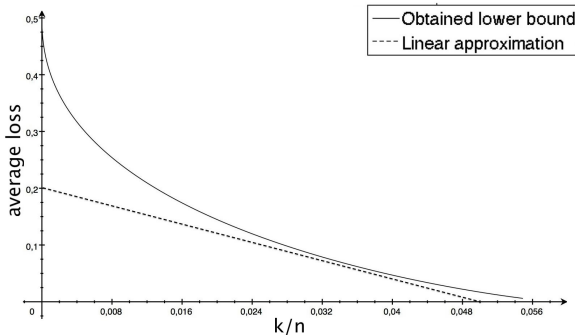
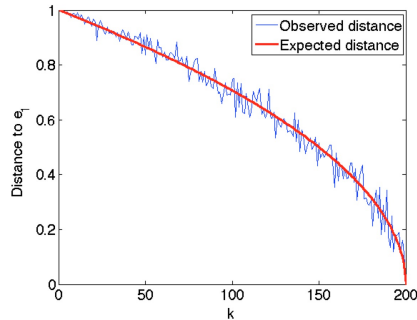

Figure 4: Lower bound for average error. The solid line is obtained by solving inequality (1). The dashed line is a simple linear bound.

Figure 5: We plot the distance of the unit vector to a subspace formed by $k$ randomly chosen instances.

linear bound of $4(\frac{1}{20} - \frac{k}{n}) = \frac{1}{5} - 4\frac{k}{n}$, because it satisfies the strict inequality for $\delta = 0.005$. It is easy to estimate, that this bound will hold for $n = 40$ with probability approximately $0.996$, and for larger $n$ that probability converges to $1$ even faster than exponentially. It remains to observe that each sign error incurs at least loss $C$, which gives us the desired bound for the average loss of the algorithm. □

The technique used in our proof also gives an interesting insight into the rigidity of random matrices. Typically, the rigidity $R_{\mathbf{M}}(r)$ of a matrix $\mathbf{M}$ is defined as the minimum number of entries that need to be changed to reduce the rank of $\mathbf{M}$ to $r$. In [FS06], a different rigidity measure, $\widetilde{R}_{\mathbf{M}}(r)$, is considered, which only counts the sign-non-preserving changes. The bounds shown there depend on the SVD spectrum of a matrix. However, if we consider a random matrix, then a much stronger lower bound can be obtained with high probability:

**Corollary 4** *For a random matrix $M \in \{-1, 1\}^{n \times n}$ and $0 < r < n$, almost certainly the minimum number of sign-non-preserving changes to a matrix in $\mathbb{R}^{n \times n}$ that is needed to reduce the rank of the matrix to $r$ is at least*

$$\widetilde{R}_{\mathbf{M}}(r) \geq \frac{n^2}{5} - 4rn.$$

Note that the rigidity bound given in [FS06] also applies to our problem, if we use the Hadamard matrix as the problem matrix. In this case, the lower bound is much weaker and no longer linear. Notably, it implies that at least $\sqrt{n}$ instances are needed to get the average loss down to zero (and this is conjectured to be tight for Hadamard matrices). In contrast our lower bound for random matrices assures that $\Omega(n)$ instances are required to get the average loss down to zero.

## 4 Random features

In this section, we argue that the family of algorithms whose weight vector is a linear combination of the instances gets confused by random features. Assume we have $n$ instances that are labeled by a single $\pm 1$ feature. We represent this feature as a single column. Now, we add random additional features. For the sake of concreteness, we add $n - 1$ of them. So our learning problem is again described by an $n$ dimensional square matrix: The $n$ rows are the instances and the target is the unit vector $\mathbf{e}_1$. In Figure 5, we plot the average distance of the vector $\mathbf{e}_1$ to the subspace formed by a subset of $k$ instances. This is the closest a linear combination of the $k$ instances can get to the target. We show experimentally, that this distance is $\sqrt{1 - \frac{k}{n}}$ on average. This means, that the target $\mathbf{e}_1$ cannot be expressed by linear combinations of instances until essentially all instances are seen (i.e. $k$ is close to $n$).

It is also very important to understand that expanding the instances using a feature map can be costly because a few random features may be expanded into many "weakly random" features that are still random enough to confuse the family of algorithms that predict with linear combination of instances. For example, using a polynomial kernel, $n$ random features may be expanded to $n^d$ features and now the sample complexity grows with $n^d$ instead of $n$.

## 5 Open problems regarding neural networks

We believe that our hardness results for picking single features out of random vectors carry over to feed forward neural nets provided that they are trained with Gradient Descent (Backpropagation) regularized with the squared Euclidean distance (Weight Decay). More precisely, we conjecture that if we restrict ourself to Gradient Descent with squared Euclidean distance regularization, then additional layers cannot improve the average loss on the problem described in Figure 1 and the bounds from Theorem 3 still hold.

On the other hand if 1-norm regularization is used, then Gradient Descent behaves more like the Exponentiated Gradient algorithm and the hardness result can be avoided.

One can view the feature vectors arriving at the output node as an expansion of the input instances. Our lower bounds already hold for fixed expansions (i.e. the same expansion must be used for all targets). In the neural net setting the expansion arriving at the output node is adjusted during training and our techniques for proving hardness results fail in this case. However, we conjecture that the features learned from the $k$ training examples cannot help to improve its average performance, provided its training algorithm is based on the Gradient Descent or Weight Decay heuristic.

Note that our conjecture is not fully specified: what initialization is used, which transfer functions, are there bias terms, etc. We believe that the conjecture is robust to many of those details. We have tested our conjecture on neural nets with various numbers of layers and standard transfer functions (including the rectifier function). Also in our experiments, the dropout heuristic [HSK$^+$12] did not improve the average loss. However at this point we have only experimental evidence which will always be insufficient to prove such a conjecture.

It is also an interesting question to study whether random features can confuse a feed forward neural net that is trained with Gradient Descent. Additional layers may hurt such training algorithms when some random features are in the input. We conjecture that any such algorithm requires at least $O(1)$ additional examples per random redundant feature to achieve the same average accuracy.

## Footnotes

[1]Since the sample space is so small it is cleaner to require small average loss on all $n$ instances than just the $n - k$ test instances. See [WV05] for a discussion.

[2]Our setup is the same as the one used in [WV05], where such hardness results were proved for the square loss only. The generalization to the more general losses is non-trivial.

[3]This is a simplification because it ignores the normalization.

[4]This can often be done efficiently via a kernel function. Our result only requires that the dot products between the expanded instances are finite and the $\phi$ map can be defined implicitly via a kernel function.

[5]The number of features $m$ can even be infinite as long as the $n^2$ dot products $\mathbf{Z}\,\mathbf{Z}^\top$ between the expanded instances are all finite. On the other hand, $m$ can also be less than $n$.

[6]Note that they count $\{-1,0,1\}$ sign patterns. However by mapping 0's to 1's we do not increase the number of sign patterns.

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
