[Reviews · NeurIPS 2014]

Submitted by Assigned_Reviewer_43

This paper considers the problem of empirical risk minimization with squared distance regularization, which results in a weight vector that is a linear combination of the training examples. The authors prove a linear lower bound on the average square loss of the algorithm on random problems, provided the loss function is nice enough, while the same problem is easy to learn by another algorithm.

This is a well-written paper on a simple idea and result, with a rather interesting interpretation. The proposed conjectures on random features and neural networks should be fleshed out in more detail, or at least with more empirical evidence.

Some additional comments/questions:

* Why does the problem matrix M (before feature mapping) need to be a square matrix?

* What can be said for an infinite-dimensional feature mapping \phi?

* L085: n/2^i --> n/2^k?

* L133: extend --> extent

* L161: ``wipes out'' is repeated
Summary: The authors show that algorithms that predict using linear combination of training examples incur at least linear loss over a random class of problems, which can be efficiently learned using another algorithm. The paper is clear and well-written, and the simple idea and result are sufficiently interesting.

Submitted by Assigned_Reviewer_45

Warmuth-Vishwanathan’05 showed that the class of kernelizable learning algorithms have some important limitations; in particular, they provided an example of a problem based on learning a number of target functions where even after observing k examples, the squared loss of _any_ kernelizable algorithm would be 1 - k/n. This paper extends the results of Warmuth-Vishwanathan’05 to different kinds of losses — logistic and hinge, in addition to the square loss.

It turns out that the example provided in the VW05 paper is not a bad example for these new kinds of losses; as a consequence, this paper provides a new example that involves a random matrix. Moreover, a completely novel analysis is employed to show that this example is a bad example for these new losses.

The problem is quite interesting, and establishes the limitations of a very heavily used class of algorithms. Although the results may seem like an extension of VW05, the analysis is quite novel, and actually rather non-trivial. As a result, I would recommend acceptance.

The presentation is quite good, but I do have a comment. The role of L2 regularization (and the relationship to WV05) seems unclear in the first two pages (although it is made clear in the related work section). I think it would help the reader to clarify this earlier in the paper.
Summary: This is an interesting paper. Although the results are an extension of Vishwanathan-Warmuth'05, the analysis is novel and extremely non-trivial. As a result, I recommend acceptance.

Submitted by Assigned_Reviewer_46

This paper extends work of [WV05] in an interesting and I think significant way. The paper considers the following problem: suppose you have n examples and a class of n possible target functions, where we fill out the matrix M, M_ij = c_j(x_i), with random {-1,+1} values. We now want to learn from a small sample. A direct counting argument (as in [BEHW87]) says that O(log n) examples are enough to learn. But suppose we require the algorithm to choose some embedding of data into R^n and then to use a linear predictor w where w is a linear combination of training examples (it must lie in the span). The work of [WV05] showed that Omega(n) training examples would be needed to get low error under square loss. This work extends that to talk about any "C-regular" loss function, which just means that if the true label is 1 and you predict a negative quantity, your loss is at least C (and similarly if the true label is -1 and you predict positive). Pretty much all reasonable loss functions are 1-regular. Interestingly, these lower bounds are stronger when M is random than if M is the Hadamard matrix. Also, the statement of the main result (Theorem 3) ends up being very clean.

Overall, the paper is in some ways a little "light" in that it essentially has one clean theorem and proof (as opposed to a collection of results and implications) but I like such papers myself. And I think the one main result is a nice one.

Question for the authors: is there any connection between your results and work showing that data of this type has no embedding where each target is a large margin separator? E.g., to take a specific example I am most familiar with: the Balcan-Blum-Srebro paper "improved guarantees for learning via similarity functions" contains a result showing that for a Hadamard matrix, for any kernel there must exist some column/target j such that any LTF has high hinge loss at any margin significantly larger than 1/sqrt(n). The analysis is based on (1) these targets are pairwise uncorrelated and therefore this class has high SQ-dimension, and (2) if there *was* a good kernel, then a *random* LTF would have a non-negligible chance of being a weak learner, violating the SQ lower bound.
Summary: A nice contribution, extending [WV05] in an interesting way to general loss functions.
Author Feedback
Author rebuttal: Thanks to all three reviewers for their positive
comments. Answers to the main questions:

Review 1:
- We will add some experimental evidence, i.e.
a plot of the generalization error of neural nets
with various number of layers.
- The problem matrix does not have to be square.
We just did this for the sake of simplicity.
We will discuss this in a footnote.
- Yes - we believe that the theorem holds for infinite
dimensional phi maps as well. However we have not formally
proven this generalization (yet). We believe that the open
problem for the finite dimensional case is more important
at this point.

Review 2:
- Good suggestion to clarify the relationship to [WV05] earlier in the intro.
We will do that.

Review 3:
- The reviewer asked a subtle question about the
relationship of our work and the known result that
certain problems cannot be embedded with a large 2-norm margin.

In short, an embedding with large margins gives good
generalization bounds. This means that if a problem cannot be
embedded with a large margin, then the generalization
bounds based on the margin argument are weak. However we don't know of
hardness results for the kernel based algorithms in terms of a
margin argument, ie. hardness (=lower bounds) of generalization for this
class of algorithms that is based on non-embeddability with
large 2-norm margins.

Curiously enough our Definition 1 is based on the number of
wrong signs instead of the size of the margin.

We will add a reference of the Balcan-Blum-Srebro paper and
an elaboration of the above arguments.